# Deciphering the Olfactory Mechanisms of *Sitotroga cerealella* Olivier (Lepidoptera: *Gelechiidae*): Insights from Transcriptome Analysis and Molecular Docking

**DOI:** 10.3390/insects16050461

**Published:** 2025-04-27

**Authors:** Hui Li, Sheng Qiao, Xiwen Hong, Yangyang Wei

**Affiliations:** 1School of Food and Strategic Reserves, Henan University of Technology, Zhengzhou 450001, China; xwhong1229@163.com; 2College of Biological Engineering, Henan University of Technology, Zhengzhou 450001, China; kyla1127@163.com; 3School of International Education, Henan University of Technology, Zhengzhou 450001, China; 18336538894@163.com

**Keywords:** *Sitotroga cerealella*, olfactory proteins, transcriptome, expression profile, molecular docking

## Abstract

This study explores the olfactory mechanisms of *Sitotroga cerealella*, a globally invasive storage pest, by analyzing its antennal transcriptome. We identified 165 presumptive olfactory genes, including odorant-binding proteins, chemosensory proteins, odorant receptors, ionotropic receptors, gustatory receptors, and sensory neuron membrane proteins. These genes show high similarity to orthologs in other insects. qRT-PCR revealed that ScerOBP15 and ScerOBP23 are highly expressed in antennae, particularly in males. Molecular docking showed strong binding affinities of ScerOBP15 to n-heptadecane and ScerOBP23 to geranyl acetone, suggesting roles in host localization. Our findings enhance the understanding of *S. cerealella*’s olfactory processes and provide a foundation for developing targeted pest management strategies.

## 1. Introduction

Insects exhibit a diverse array of behaviors, including courtship, kin recognition, aggregation, foraging for food, and host-plant recognition, all of which are mediated by information from their olfactory and chemosensory systems [1]. Olfaction constitutes a complex network comprising various proteins, including odorant-binding proteins (OBPs), chemosensory proteins (CSPs), odorant receptors (ORs), ionotropic receptors (IRs), gustatory receptors (GRs), sensory neuron membrane proteins (SNMPs), and odorant-degrading enzymes (ODEs) [2]. The olfactory process initiates when odorant molecules diffuse through pores into the sensillum lymph, where they bind to small, amphipathic proteins, specifically OBPs and CSPs. Subsequently, OBPs or CSPs transport the odorant molecules through the sensillum lymph to the ORs, activating the olfactory receptor neurons (ORNs) and converting chemical signals into electrical impulses that are transmitted to the insect brain [3,4,5]. This intricate process underscores the sophisticated nature of the insect olfactory system and its critical role in mediating essential behaviors.

OBPs, recognized as critical transporters of odorant molecules and pheromones, are specifically concentrated within the sensilla lymph of insect antennae [6,7]. The first OBP, designated as pheromone-binding protein, was identified in the antenna of *Antheraea polyphemus* [8]. Since then, these small (100–300 amino acids) water-soluble globular proteins have been characterized across various insect species through antennal transcriptome analyses [9,10]. These OBPs are classified into several subgroups based on the number of conserved cysteine residues: classic OBPs with six conserved cysteines, Minus-C OBPs with four, Plus-C OBPs with eight, and Dimer OBPs with twelve [10,11]. Ligand-binding assays have demonstrated that pheromone-binding proteins (PBPs) selectively bind pheromones, while general odorant-binding proteins (GOBPs) interact with a broad spectrum of odorants from host plants [12,13]. Similar to OBPs, CSPs are small soluble proteins characterized by four conserved cysteine residues and exhibit greater conservation than OBPs. They are found in the antennal sensillum lymph, as well as in other non-chemosensory organs, suggesting additional roles beyond the transport of odorant molecules [14,15,16].

In insects, olfactory reception is mediated by two families of olfactory receptors, namely odorant receptors (ORs) and ionotropic receptors (IRs), along with one family of gustatory receptors (GRs). Classical ORs are characterized by the presence of seven conserved transmembrane domains and were first identified in the genome of *Drosophila melanogaster* [14,17]. Pheromone molecules are typically recognized by specific pheromone receptors (PRs). Additionally, insect ORs, in conjunction with the olfactory co-receptor Orco, form an odor-gated ion channel complex that converts chemical signals into electrical signals [16,18,19]. The IR family, first characterized in *D. melanogaster*, possesses a molecular configuration analogous to ionotropic glutamate receptors [20,21,22]. Compared to ORs, insect IRs are implicated in the detection of acids, amines, and aldehydes within the olfactory system, as well as in various non-chemosensory functions [23,24]. The GR family, expressed in the antennae, proboscis, and palps of insects, plays a crucial role in the detection of CO_2_, sugars, and bitter substances [25,26,27,28].

Sensory neuron membrane proteins (SNMPs), specifically SNMP1 and SNMP2, are found in the antennae of insects. SNMP1 co-localizes with pheromone receptors in pheromone-specific olfactory sensory neurons (OSNs), whereas SNMP2 is found in the support cells of the sensilla [29,30,31]. These proteins are hypothesized to play an essential role in pheromone perception.

Molecular docking and molecular dynamics simulation are the leading techniques for probing the interactions between olfactory proteins and ligands [32,33]. A foundational step in molecular docking is obtaining the structural model of the target protein. In the absence of experimental protein structures, homology modeling uses the tertiary structures of closely related OBPs as templates to predict the three-dimensional structure of OBPs [34]. To date, the three-dimensional structures of over 20 OBPs have been characterized across diverse insect orders, including Lepidoptera, Diptera, and Hemiptera [35,36]. These techniques are widely used to analyze the binding profiles between olfactory proteins and ligands in insects and to compare the binding profiles among different proteins [37,38]. Molecular docking has been applied in virtual screening for insect behavior regulators across various insect orders, such as Coleoptera, Diptera, Hemiptera, Hymenoptera, Lepidoptera, Orthoptera, and Trichoptera [39]. This approach enables the large-scale screening of lead compounds for insect behavior regulators, significantly accelerating their development. It provides a vital theoretical basis for studies on olfactory protein-binding affinities and has substantial implications for the future discovery of novel behavior regulators [37,40].

The Angoumois grain moth, *Sitotroga cerealella* (Olivier) (Lepidoptera: *Gelechiidae*), is a pervasive pest that poses a substantial threat to the storage of grains, such as wheat, rice, and maize. Historically, phosphine fumigation has been the predominant control method. However, a prolonged reliance on pesticides can lead to residues, resistance, and environmental pollution. Recently, modulating olfactory behavior has risen as a promising strategy for pest management. For instance, diallyl trisulfide, a component of garlic, has been shown to disrupt the binding of sex pheromones to PBPs [41]. Thus, elucidating the molecular basis of insect olfaction is essential for devising innovative bio-pesticides and integrated pest management strategies [42]. Despite this, the molecular mechanisms of chemical communication in *S. cerealella* remain largely elusive.

In this study, we performed transcriptome analyses on the antennae of both mated and unmated female and male adults of *S. cerealella*, identifying 165 putative olfactory genes, comprising 33 OBPs, 10 CSPs, 58 ORs, 41 IRs, 21 GRs, and 2 SNMPs. We then conducted sequence alignment and a phylogenetic analysis of these olfactory genes. Furthermore, we evaluated the transcript expression levels of these candidate olfactory genes across various tissues in female and male adults using transcriptome data. The binding affinities of ScerOBP15 and ScerOBP23 to specific wheat volatiles were preliminarily predicted using three-dimensional structural modeling and molecular docking. This research establishes a foundation for future functional studies on these olfactory genes in *S. cerealella* at the molecular level.

## 2. Materials and Methods

### 2.1. Insect Rearing and Tissue Collection

The adults of *S. cerealella* were sourced from Zhengzhou (112°42′ E, 34°16′ N), Henan Province, China, and were reared on wheat in 4 L plastic containers under stringent environmental conditions: 28 ± 2 °C, 70 ± 5% relative humidity, and a dark photoperiod. Antennae (At) and bodies with antennae removed (both mated and unmated females and males) were dissected and preserved in 1.5 mL Eppendorf tubes with RNAlater (Ambion, Austin, TX, USA). Thereafter, the samples were rapidly frozen in liquid nitrogen and stored at −80 °C for subsequent analysis.

### 2.2. RNA-Seq Library Construction and Illumina Sequencing

Total RNA extraction from the antennae of mated and unmated female and male *S. cerealella* was performed using TRIzol reagent (Invitrogen, Carlsbad, CA, USA) following the manufacturer’s guidelines. The RNA concentration and purity were determined using a NanoDrop spectrophotometer, and RNA integrity was validated through electrophoresis on a 1.0% agarose gel and subsequent analysis with an Agilent 2100 Bioanalyzer (Agilent Technologies, Santa Clara, CA, USA). Poly(A) mRNA was purified and enriched from the total RNA using oligo(dT) magnetic beads, fragmented, and subjected to first-strand cDNA synthesis with random hexamer primers. Second-strand cDNA synthesis involved RNase H, buffer, dNTPs, and DNA polymerase I. The double-stranded cDNA underwent end-repair and modification with an A-tailing at the 3′ end and phosphorylation at the 5′ end. Subsequently, PCR amplification generated the library, which was then assessed for quality using the Agilent 2100 Bioanalyzer (Agilent Technologies, Santa Clara, CA, USA). Ultimately, the libraries were sequenced on the MGISEQ-2000 platform (BGI-Shenzhen, China).

### 2.3. De Novo Assembly and Sequence Annotation

Clean reads for de novo assembly were obtained by removing the adapters and low-quality sequences from the raw reads with SOAPnuke (v2.1.0) [43]. Assembly was conducted using Trinity software (v2.4.0) to produce nonredundant unigenes [43]. Functional annotation of nonredundant unigenes from the *S. cerealella* antennal transcriptome was performed using Diamond blastx against public protein databases: the NCBI non-redundant (NR, e-value < 10^−5^) database, Gene Ontology (GO, e-value < 10^−5^), Swiss-Prot (e-value < 10^−5^), euKaryotic Orthologous Groups/clusters of Orthologous Groups of programs (KOG/COG, e-value < 10^−3^), and the Kyoto Encyclopedia of Genes and Genomes (KEGG, e-value < 10^−10^) [44].

### 2.4. Identification of Olfactory Genes and Phylogenetic Analysis

Candidate olfactory genes in *S. cerealella* were identified through functional annotation results from five publicly available protein databases and validated against the NCBI protein database using BLASTx (https://blast.ncbi.nlm.nih.gov/Blast.cgi, accessed on 23 May 2023). Open reading frames (ORFs) of putative olfactory genes were predicted with the ORF finder (http://www.ncbi.nlm.nih.gov/gorf/gorf.html, accessed on 23 May 2023), and signal peptides were predicted with the SignalP 4.0 program (http://www.cbs.dtu.dk/services/SignalP/, accessed on 28 May 2023). The TMHMM Server 2.0 was employed to predict the transmembrane domains (TMDs) of the olfactory proteins (http://www.cbs.dtu.dk/services/TMHMM, accessed on 28 May 2023) [42].

Phylogenetic trees were constructed with amino acid sequences from *S. cerealella* and other insects’ olfactory genes using MEGA7.0 with the maximum-likelihood method, 1000 bootstrap replications, and a poison model. FigTree v.1.4.4 software and Adobe Illustrator CS6 (v5.2.0.17) were used to edit and render the trees. Redundant sequences from other insects were removed to enhance the quality of the phylogenetic trees, except for those from *S. cerealella*.

### 2.5. Expression Analysis of Olfactory Genes

The FPKM approach was employed to quantify the expression profiles of olfactory genes in the antennae of both female and male *S. cerealella* [45]. Transcriptomic differences among genes were identified using the DEGseq tool with a false discovery rate threshold of less than 0.05, and an absolute value of |log2(FoldChange)| > 1 to determine statistical significance in gene expression [46]. Comparative DEG analysis was conducted to confirm the differential expression of the chemosensory genes between unmated males and females.

Tissues from adult *S. cerealella*, including antennae (At) and body segments devoid of antennae (B), were dissected and preserved in RNase-free tubes (AXYGEN). Total RNA was extracted from these distinct tissues following the manufacturer’s protocol using TRIzol reagent (Takara, Dalian, China). cDNA synthesis was performed using the RevertAid First Strand cDNA Synthesis Kit (Thermo Scientific, Waltham, MA, USA). The expression profiles of the olfactory genes across various adult tissues were validated by qPCR on a Light Cycler 480 System (Rochester, NY, USA) with SYBR Premix EX Taq (Takara, China). A constitutive *S. cerealella* housekeeping gene, glyceraldehyde-3-phosphate dehydrogenase (GAPDH), served as an endogenous control to standardize gene expression and adjust for sample variability [41]. The primers are listed in Appendix A. Primers for the target and reference genes were designed with Primer 3 Plus (https://www.primer3plus.com/index.html, accessed on 6 July 2023), with expected amplicon sizes ranging from 100 bp to 200 bp. The qPCRs were set up in 25 μL volumes, comprising 1 μL of each primer (10 μM), 1 μL of cDNA template (200 ng/μL) from tissues such as the antennae and body, 0.375 μL of reference dye (50×), 12.5 μL of 2×SYBR Green QPCR Master Mix (Stratagene, La Jolla, CA, USA), and 9.125 μL sterilized ultrapure H_2_O. The thermal-cycling profile consisted of 95 °C for 3 min, followed by 40 cycles of 10 s at 95 °C and 30 s at 55 °C. Dissociation curves were generated by heating the PCR products to 95 °C for 15 s, cooling to 57 °C for 15 s, and then heating to 95 °C for 15 s, with this cycle repeated for 20 min. Each RT-qPCR assay was conducted in triplicate per sample for technical replicates. No-template and no-transcriptase controls were included in each qPCR run to verify specificity. To ensure reproducibility, each qPCR was independently repeated three times for each sample.

The relative quantification 2^−ΔΔCT^ approach was applied to evaluate the relative transcript abundance of genes in each tissue (where ΔΔCT = (CTScerOBP − CTScerGAPDH rRNA) Time x − (CTScerOBP – CTScerGAPDH rRNA) Time 0) [47].

### 2.6. Structure Modeling and Molecular Docking of Ligands

The three-dimensional structure of ScerOBP15 and ScerOBP23 was predicted using the SWISS-model portal at the Expasy tools platform (https://swissmodel.expasy.org/interactive, accessed on 25 July 2023). Assessment of the 3D model quality, including protein geometry, was conducted with UCLA-DOE LAB-SAVES v6.0 (https://saves.mbi.ucla.edu/, accessed on 12 August 2023) [48,49,50]. Molecular docking studies employed the Auto Dock Vina 1.2.3 software [51,52]. The newly constructed 3D model of ScerOBP15 and ScerOBP23 served as receptors, and ligands were selected from PubChem based on significant experimental affinities. Default settings were applied during the Auto Dock Vina 1.2.3 molecular-docking process, and complexes were identified based on the lowest binding energy. Structural analysis of the visual and docked complexes was performed using the Protein–Ligand Interaction Profiler (https://plip-tool.biotec.tu-dresden.de/plip-web/plip/index, accessed on 27 August 2023) and the PyMOL tool (v2.5.7) [53,54].

## 3. Results

### 3.1. Transcriptome Sequencing, Assembly, and Annotation

The antennal transcriptome of unmated male *S. cerealella* produced a total of 25.33 million (SE ± 4.61 million) high-quality reads, achieving a Q20 score of 98.23% (±0.16). The mated male counterpart yielded 23.75 million (SE ± 1.36 million) high-quality reads, also with a Q20 of 98.23% (±0.06). Unmated females produced 26.65 million (SE ± 1.95 million) high-quality reads with a Q20 of 98.23% (±0.16), while mated females generated 26.64 million (SE ± 1.27 million) clean reads, also scoring 98.23% (±0.20) in Q20 (Table 1). Assembly of the *S. cerealella* antennal transcriptome resulted in 142,382 transcripts, with 47,663 (33.48%) being shorter than 500 bp and 33,404 (23.46%) exceeding 2000 bp in length (Table 1). The raw reads from the *S. cerealella* antennal transcriptome have been submitted to the NCBI Short Read Archive (SRA) database under accession number PRJNA877069.

A total of 67,753 (47.59%) transcripts showed sequence similarity to known entries within at least one protein database. Among these, 30,846 (21.67%) genes were classified into 26 KOG categories, with the ‘General function prediction only’ category being the most abundant (Figure 1A). An analysis indicated that 17,666 (12.41%) transcripts were associated with three GO categories: biological process, cellular component, and molecular function (Figure 1B). Additionally, 34,591 (24.29%) genes were mapped to five KEGG pathways, encompassing 34 distinct groups (Figure 1C).

### 3.2. Olfactory-Related Genes in S. cerealella

Following BLAST analysis and functional annotation against five protein databases, we identified a total of 167 putative olfactory genes from the antennal transcriptomes of *S. cerealella*. This set includes 33 OBPs, 10 CSPs, 58 ORs, 41 IRs, 21 GRs, 2 SNMPs, and 2 ODEs.

#### 3.2.1. Identification and Analysis of Putative OBPs

In this investigation, 33 OBPs from the antennal transcriptome of *S. cerealella* were identified, which included 4 PBPs, 3 GOBPs, 1 antennal-binding protein (ABP), and 25 classic OBPs. Three OBP-encoding genes, ScerPBP2, ScerGOBP1, and ScerGOBP2, have been previously reported. Sequence analysis revealed that 18 ScerOBPs had complete ORFs, and 21 ScerOBPs possessed signal peptide sequences. BLAST comparisons indicated that all OBPs exhibited over 51% amino acid sequence identity with their Lepidopteran orthologs, with ScerOBP20 displaying over 96% identity with GOBPs from *Amyelois transitella*. Multiple sequence alignment results demonstrated that 17 ScerOBPs harbored the conserved motif, while 5 ScerOBPs were classified as Minus-C OBPs. A phylogenetic analysis revealed that all ScerPBPs (ScerPBP1–4) grouped with PBPs from other Lepidoptera, and two ScerGOBPs (ScerGOBP1 and ScerGOBP2) fell within the GOBPs clade, whereas ScerGOBP3 clustered with ABPs and OBPs from other Lepidoptera (Figure 2).

#### 3.2.2. Identification and Analysis of Putative CSPs

A total of 10 CSPs were identified in this study. A sequence analysis revealed that nine ScerCSPs possessed complete ORFs with polypeptide lengths varying from 103 to 158 amino acid residues. Additionally, all identified sequences featured an N-terminal signal peptide. Multiple sequence alignment indicated that all ScerCSPs contained a conserved motif with four conserved cysteine residues. A phylogenetic analysis of the ScerCSPs demonstrated the presence of three distinct clades (clades I, II, and III), which is consistent with the patterns observed in other Lepidopteran species. Specifically, ScerCSP6 was found to be monophyletic with CpinCSP16 in clade I, while two ScerCSP transcripts (ScerCSP2 and ScerCSP9) were associated with the divergent clade II. Clade III was monophyletic, with a large Lepidoptera-specific clade encompassing the remaining seven ScerCSPs (Figure 3).

#### 3.2.3. Identification and Analysis of Putative ORs

A total of 58 candidate ORs from the antennal transcriptomes of *S. cerealella* were identified based on BLASTx annotation results. Among these, 26 ScerORs possessed complete ORFs, encoding proteins with lengths ranging from 162 to 474 amino acids. Furthermore, three ScerORs (ScerOrco, ScerOR31, ScerOR15) were characterized by the presence of seven transmembrane domains (TMDs). A BLASTx analysis revealed that the majority of ScerORs exhibited low sequence similarity (31.82–78.26%) with orthologs from other insect species, with the exception of ScerOrco. A phylogenetic analysis confirmed that one OR, named ScerOrco, is conserved within the Orco gene family, closely related to SinsOrco and GmelORco. Additionally, all ScerPRs (ScerPR1–4) were found to be part of the pheromone receptor (PR) family (Figure 4).

#### 3.2.4. Identification and Analysis of Putative IRs

A bioinformatic analysis of the *S. cerealella* antennal transcriptome identified a total of 41 candidate IRs, with 18 ScerIRs exhibiting complete ORFs that spanned 473 to 916 amino acid residues. A phylogenetic analysis, incorporating ScerIRs and IRs from other Lepidopteran insects, revealed that ScerIRs segregated into 11 distinct groups, encompassing IR8a, IR25a, IR21a, IR40a, IR41a, IR64a, IR68a, IR75d, IR75q, IR75p, and IR93a. Notably, ScerIR8a is grouped within the IR8a clade, hypothesized to function as co-receptors with other IRs, and demonstrated a high sequence identity of 98.85% with IR8a from *Athetis lepigone* (Figure 5).

#### 3.2.5. Identification and Analysis of Putative GRs

In the antennal transcriptome of *S. cerealella*, we identified 22 candidate gustatory receptors (ScerGRs 1–22), with 10 ScerGRs exhibiting incomplete ORFs due to the absence of either a 5′ or 3′ terminus. Of these, nine genes were nearly full-length, possessing complete ORFs that encoded proteins with lengths ranging from 326 to 483 amino acids and featuring 6–8 transmembrane domains. A phylogenetic analysis revealed that all ScerGRs were segregated into four distinct clades: ‘CO_2_ receptors’, ‘sugar receptors’, ‘fructose receptors’, and ‘bitter receptors’. ScerGR14–16 were classified within the ‘CO_2_ receptors’ clade, alongside BmorGR1N, BmorGR2NJ, and BmorGR3F, which are known carbon dioxide sensors. Seven ScerGRs were found to cluster with sugar-sensing receptors (BmorGR4S, 5NJF, 6P, 7, 8R) from Bombyx mori. Additionally, ScerGR19 and ScerGR20 were grouped with fructose receptors, including BmorGR9R and BmorGR10. The remaining nine ScerGRs were categorized under the ‘bitter receptors’ clade (Figure 6).

#### 3.2.6. Identification and Analysis of Putative SNMPs

In the transcriptomes of adult *S. cerealella*, we identified two candidate sensory neuron membrane proteins (SNMPs). Both proteins possessed ORFs, with lengths varying from 521 to 524 amino acid residues, and featured two transmembrane domains (TMDs). A BLASTx analysis indicated that ScerSNMP1 exhibited 82.03% sequence similarity to HkahSNMP1, while ScerSNMP2 showed 71.32% sequence identity to HkahSNMP2. A phylogenetic analysis revealed three distinct groups, with ScerSNMP1 clustering within the SNMP1 clade and ScerSNMP2 within the SNMP2 clade, alongside other Lepidopteran SNMP2s. Notably, both the SNMP1 and SNMP2 clades were prevalent in the antennae of the insects (Figure 7).

### 3.3. Identification of Sex-Bias Expression Genes Based on Transcriptome

Analysis of the FPKM values derived from the *S. cerealella* antennal transcriptome (Figure 8) revealed that five OBPs, including ScerGOBP2, ScerPBP1, ScerPBP3, ScerOBP15, and ScerOBP23, were markedly expressed in male antennae (FPKM > 10,000). In contrast, two OBPs, ScerGOBP1 and ScerGOBP2, exhibited high expression in female antennae (FPKM > 10,000). ScerCSP8 was identified as the predominant CSP in the antennae of *S. cerealella*. ScerOrco and ScerIR25a were the most abundantly expressed OR and IR, respectively, potentially reflecting their specialized roles. ScerGR19 was the predominant GR in the antennae of *S. cerealella*, and ScerSNMP1 was the most highly expressed SNMP. A total of 25 olfactory-related genes, comprising 11 ScerOBPs, 11 ScerORs, 2 ScerSNMPs, and 1 ScerCSP, demonstrated sex-biased expression in mated *S. cerealella*, with a male bias. In virginal *S. cerealella*, 26 olfactory-related genes (11 ScerOBPs, 12 ScerORs, 2 ScerSNMPs, and 1 ScerCSP) showed sex-biased expression, with 23 genes exhibiting a male bias and the remaining 3 showing a female bias.

A semi-quantitative reverse transcription quantitative PCR (RT-qPCR) was employed to investigate the expression profiles of ScerOBP15 and ScerOBP23 across the adult tissues of *S. cerealella*, encompassing antennae and bodies devoid of antennae. The RT-qPCR findings revealed significantly elevated expression levels of ScerOBP15 and ScerOBP23 in male antennae relative to other tissues (*p* < 0.01; Figure 9). Additionally, there was no significant variation in the expression levels of ScerOBP15 and ScerOBP23 among the female antennae, female bodies, and male bodies (Figure 9).

### 3.4. Protein Structure and Interaction Analysis by 3-Dimensional Docking

To elucidate the interactions between ScerOBP15, ScerOBP23, and the volatiles from the wheat and *S. cerealella*, we modeled the 3D structures of ScerOBP15 and ScerOBP23 using the Swiss-model software. The template details are presented in Table 2. The accuracy of the constructed models was evaluated using ERRAT, Verify-3D, and ProCheck, with the outcomes indicating that the models’ structures are plausible (Appendix A). Sequence alignment of the target proteins with their respective templates was conducted using the ESPript 3.0 tool [55]. The analysis revealed that both ScerOBP15 and ScerOBP23 possess six conserved cysteine residues, which can form three disulfide bonds (Appendix A). Additionally, both proteins feature six α-helices, which is consistent with the typical structural features of OBPs. We accordingly reasoned that the 3D structures of ScerOBP15 and ScerOBP23 were acceptable and could be used for further analysis.

An analysis reveals that the binding affinity of ScerOBP15 for 12 ligands spans from −6.201 to −4.406 kcal/mol, as outlined in Table 3. Notably, ScerOBP15 exhibits the strongest binding affinity with geranyl acetone, with a binding energy of −6.201 kcal/mol. Following is oleic acid, with a binding energy of −6.016 kcal/mol, and the interaction with pentadecanoic acid is characterized by a binding energy of −5.721 kcal/mol. The binding energy with eight other volatiles, substantial at −5.00 kcal/mol, indicates a high binding affinity for these substances. Docking experiments conducted with the ScerOBP15 protein model identified several key binding sites (Table 3). Figure 10 illustrates the three-dimensional binding configurations, binding cavities, and two-dimensional interaction diagrams of the docked complexes. Key residues within the ScerOBP15 binding pocket, such as IEU-4, ILE-44, LEU-48, LEU-67, TRP-104, PHE-113, and LEU-114, are primarily involved in ligand interactions, predominantly through hydrophobic interactions.

Molecular-docking analyses revealed that ScerOBP23 exhibits binding affinities of −5.938, −5.907, −5.811, and −5.632 kJ/mol with n-heptadecane, oleic acid, geranyl acetone, and pentadecanoic acid, respectively. This indicates that ScerOBP23 is capable of forming stable complexes with these four ligand molecules. The spatial architecture of the residues and the nature of the interaction forces are pivotal for understanding the key interactions between these amino acid residues and ligands. Figure 11 presents the three-dimensional binding configurations, binding cavities, and two-dimensional interaction diagrams of the docked complexes. Key residues within the ScerOBP23 binding pocket, including ILE-15, IEU-54, IEU-55, ALA-58, LEU-60, LEU-72, LEU-75, ILE-76, VAL-84, VAL-119, and PHE-121, are engaged in ligand interactions. In the n-heptadecane–ScerOBP23 complex, these residues primarily exhibit hydrophobic interactions, with odor molecules largely enveloped by hydrophobic amino acid-rich binding cavities.

## 4. Discussion

In recent years, advancements in high-throughput sequencing technologies have facilitated the identification of numerous olfactory-related genes in various insect species, including *Oxya chinensis* (Thunberg) [56], *Cotesia vestalis* (Haliday) [57], *Locusta migratoria* (L.) [58], *Micromelalopha troglodyta* (Graeser) [59], *Anastatus japonicus* (Ashmead) [60], and *Ceracris nigricornis* (Walker) [42]. Despite these findings, a systematic characterization of the olfactory-related genes in *S. cerealella* remains lacking. To address this gap, we utilized the MGISEQ-2000 platform for sequencing the antennal transcriptome of *S. cerealella*, resulting in the identification of 167 olfactory-related genes, encompassing 33 OBPs, 10 CSPs, 58 ORs, 41 IRs, 21 GRs, 2 SNMPs, and 2 ODEs. Subsequently, we explored the expression profiles, ligand affinities, and binding mechanisms of ScerOBP15 and ScerOBP23. Our mining of olfactory-related genes aims to elucidate the olfactory mechanisms of *S. cerealella* and lays the groundwork for developing eco-friendly control strategies.

Odorant-binding proteins are responsible for transporting odor molecules to olfactory receptor neurons (ORNs), where they activate olfactory receptors (ORs) or ionotropic receptors (IRs), ultimately conveying the signals to downstream effectors within the olfactory system [56]. The antennal transcriptome of *S. cerealella* revealed a number of ScerOBPs comparable to certain species, with 28 identified in *S. insularis* and 32 in *Micromelalopha troglodyta* [16,59], yet surpassing the counts in *Oxya chinensis* (18 OBPs) [56], *Cotesia vestalis* (22 OBPs) [57], *Hylamorpha elegans* (16 OBPs) [61], and *Ceracris nigricornis* (20 OBPs) [42]. Four genes were designated as candidate pheromone-binding proteins due to their clustering with PBPs from other Lepidopteran species in the phylogenetic analysis of *S. cerealella* OBPs. An analysis of the antennal transcriptome data revealed that ScerPBPs are more highly expressed in male antennae than in female antennae, indicating a potential role in mate recognition in *S. cerealella*. Similarly, certain PBPs, such as CpunPBP2, exhibit elevated expression in male antennae, suggesting that PBP genes may serve distinct functions across sexes. Given that PBPs are known to be modulated by sex pheromone release, mating, or oviposition, these genes likely play crucial roles in reproductive behaviors [62]. Additionally, a phylogenetic analysis revealed that ScerGOBP1 formed a distinct clade with GOBPs from various Lepidopteran species, including *G. mellonella*, *M. separata*, *L. botrana*, *S. insularis*, *L. sticticalis*, and *C. pinicolalis*, while ScerGOBP2 clustered with GOBP2 from these species. This suggests a shared evolutionary origin for these GOBPs. An expression analysis indicated that ScerGOBP1 and ScerGOBP2 were highly expressed in both antennal samples, potentially implicating them in the detection of common odors. Considering that several OBPs from *Grapholita funebrana* are highly expressed in the antennae of both male and female moths, and that GfunGOBPs exhibit a dual capacity for selectively binding to sex pheromones and host-plant volatiles [63,64], it is plausible that their *S. cerealella* counterparts may possess analogous functions. However, additional research is necessary to explore the potential evolutionary conservation of these functions.

Based on FPKM values, ScerOBP15 and ScerOBP23 exhibited high levels of expression in both male and female antennae. However, analysis with the DESeq2 program indicated that the expression levels of these genes in male antennae were markedly higher than in female antennae, a finding confirmed by RT-qPCR. Additionally, RT-qPCR revealed no significant difference in the expression levels of ScerOBP15 and ScerOBP23 between male and female bodies lacking antennae. Consistent with these results, Semanotus bifasciatus exhibited similar expression patterns for SbifOBP18 and SbifOBP21 in both sexes’ antennae [65].

Within the insect olfactory system, CSPs participate in the transport of odor molecules, akin to OBPs [66,67]. CSPs, characterized by four conserved cysteine residues, exhibit greater sequence conservation compared to OBPs [68,69]. In our study, ten ScerCSPs were identified in the antennal transcriptome of *S. cerealella*, each featuring a 16–22 amino acid signal peptide. Beyond their chemoreceptive role, CSPs are implicated in a variety of functions. Numerous studies have documented the presence of CSPs in pheromone glands, suggesting their potential involvement in pheromone release. Additionally, CSPs are hypothesized to play a role in insect development and regeneration. For instance, in *Blatta germanica*, a CSP is involved in leg regeneration [70]. OcomCSP12 is specifically expressed in *Ophraella communa* ovaries, and silencing this gene will result in the embryos not developing completely [71]. A similar phenomenon is observed in Apis mellifera [72]. Furthermore, CSPs are engaged in fatty acid biosynthesis and other metabolic pathways [73].

In this research, we identified 58 ORs from the antennal transcriptome of *S. cerealella*, with 26 possessing ORFs. Notably, 12 of these ORs exhibited a significant expression difference in FPKM values. Within *S. cerealella*, ScerPR1–3 were characterized as pheromone receptors (PRs), and an FPKM analysis revealed their specific expression in male antennae. Additionally, eight other ScerORs displayed a male antennae-enriched expression pattern, though their specific functions remain elusive and necessitate further investigation. Functional annotation of the antennal transcriptome revealed the presence of an Orco homolog in *S. cerealella*, and phylogenetic analysis indicated that ScerOrco clusters within a conserved clade with Orco from other Lepidopteran species. In insects, Orco is a highly conserved protein, typically with a single copy present per species. Typically, heterotetrameric odorant-gated ion channels are formed by the association of Orco with a variable OR [18].

A total of 41 IRs were identified in the antennal transcriptome of *S. cerealella*, surpassing the numbers identified in *L. sticticalis* (17) [14], *Hylamorpha elegans* (22) [61], *C. medinalis* (13) [74], *G. mellonella* (22) [75], and *Halyomorpha halys* (24) [76]. A phylogenetic analysis, incorporating ScerIRs with IRs from other Lepidopteran species, revealed 11 distinct clusters, including the IR8a, IR25a, IR21a, IR40a, IR41a, IR64a, IR68a, IR75d, IR75q, IR75p, and IR93a groups. IR8a and IR25a are recognized as co-receptors, as evidenced by their co-expression with other IRs in prior studies [10]. The roles of IRs are more intricate than those of ORs, extending beyond olfaction to include the detection of amines, acids, taste, and temperature sensing. For instance, IR21a and IR93a have been implicated in thermosensation in *Drosophila* [77,78]. IR41a is involved in long-range attraction to the polyamines [79]; and IR64a, specifically activated by acid, is suggested to play a crucial role in acid detection in *Drosophila* [80].

In our analysis of *S. cerealella*, we identified 21 GRs, a count lower than that observed in *B. mori* (65) and *Manduca sexta* (45) [81,82]. This discrepancy could be attributed to the specific expression of certain GRs in non-olfactory tissues [83]. Numerous moth species, including *S. cerealella*, utilize GRs for the detection of sugar, fructose, CO_2_, and bitter components [11]. A phylogenetic analysis indicated that ScerGR14–16 grouped with potential CO_2_ receptor orthologs (Bmor1N, 2NJ, and 3F) from *B. mori*, implying a potential role in CO_2_ detection in *S. cerealella* [82]. Nine GRs (Scer3–5, 7, 9, 10, 12, 19, 20) were classified within the sugar and fructose receptor clades. Empirical evidence supports the crucial role of sugar and fructose in host location and oviposition behavior in female moths, suggesting that these nine GRs may modulate egg-laying behavior [83]. Additionally, nine ScerGRs were identified within the bitter receptor clade, potentially involved in the perception and discrimination of plant-derived toxic secondary metabolites [82].

ScerOBP15 and ScerOBP23 exhibit broad binding spectra for volatile compounds in wheat grains, with 8 out of 12 host volatile compounds demonstrating binding activity. Notably, they exhibit strong binding affinity for n-heptadecan, oleic acid, pentadecanoic acid, and geranylacetone. ScerOBP15 shows a stronger binding capacity to oleic acid, pentadecanoic acid, and geranyl acetone compared to ScerOBP23, but a slightly weaker affinity for n-heptadecan. This suggests that both ScerOBP15 and ScerOBP23 are capable of selectively recognizing volatile compounds from host plants. In other insects, OBPs have been shown to selectively bind to host volatile compounds or environmental volatile information substances. For instance, HparOBP14 from the Black Gill tortoise beetle can bind to the host-plant volatile compound 6-Methyl-5-hepten-2-one, as well as other volatiles such as p-Xylene, methanol, formaldehyde, α-pinene, and geraniol [84]. DcitOBP2 from the citrus mealybug binds to the host volatile compounds methyl disulfide and D-limonene [85]. SlitOBP34 from Spodoptera litura binds to host-plant volatile compounds, including benzaldehyde, 1-hexanol, and ethyl cis-3-hexenoate [86]. Crystal structure studies of OBPs using NMR have revealed that OBPs typically interact with different ligands through non-polar or polar amino acid residues in their hydrophobic binding cavities, forming hydrogen bonds, hydrophobic interactions, and van der Waals forces [87,88]. For example, AcerOBP2 forms hydrogen bonds with 4-methoxybenzaldehyde and (E)-ethyl cinnamate, while with methyl cinnamate, only hydrophobic interactions and van der Waals forces are observed [89]. This study demonstrates that ScerOBP15 and ScerOBP23 have the strongest binding effects with geranylacetone and n-heptadecan, respectively, with binding energies of −6.201 kcal/mol and −5.938 kcal/mol. Molecular-docking results indicate that geranylacetone is located within a binding cavity formed by six α-helices in ScerOBP15 and stabilizes the protein–ligand complex by forming hydrophobic interactions with seven hydrophobic amino acid residues, including IEU-4, ILE-44, LEU-48, LEU-67, TRP-104, PHE-113, and LEU-114. Similarly, n-heptadecan is situated in the binding cavity of ScerOBP23 and forms hydrophobic interactions with several surrounding hydrophobic amino acid residues, resulting in a tight binding with ScerOBP23. The binding of geranylacetone and n-heptadecan to ScerOBP15 and ScerOBP23, respectively, suggests that these ligands are enclosed by a hydrophobic cavity, leading to a strong receptor–ligand interaction. This binding mode is analogous to that of Apis cerana AcerOBP2 with methyl cinnamate, highlighting the pivotal role of hydrophobic amino acids in OBP–ligand binding [89]. This study predicts the key amino acid residues involved in odor ligand binding by ScerOBP15 and ScerOBP23 using molecular docking. However, molecular docking is often a static and instantaneous simulation, potentially deviating from actual protein-binding scenarios. Therefore, to further elucidate the specific role of these hydrophobic amino acids in the recognition of host volatiles by ScerOBP15 and ScerOBP23, additional approaches, such as site-directed mutagenesis and fluorescence competition binding assays, are required to assess the impact of key amino acid mutations on the ligand-binding capacity of ScerOBP15 and ScerOBP23.

## 5. Conclusions

The primary objective of this study was to identify the genes underlying olfactory sensation, host localization, feeding behavior, and oviposition preferences in *S. cerealella* through transcriptome sequencing analysis. We annotated a total of 167 chemoreception-related transcripts, comprising 33 OBPs, 10 CSPs, 58 ORs, 41 IRs, 21 GRs, 2 SNMPs, and 2 ODEs. Phylogenetic trees were constructed, and thorough comparative analyses were conducted. Additionally, we conducted preliminary structural and functional analyses of ScerOBP15 and ScerOBP23, focusing on the role of key residues in protein–ligand interactions. This research offers significant insights into the olfactory recognition mechanisms of *S. cerealella* and facilitates the further functional characterization of these genes.

## Figures and Tables

**Figure 1 insects-16-00461-f001:**
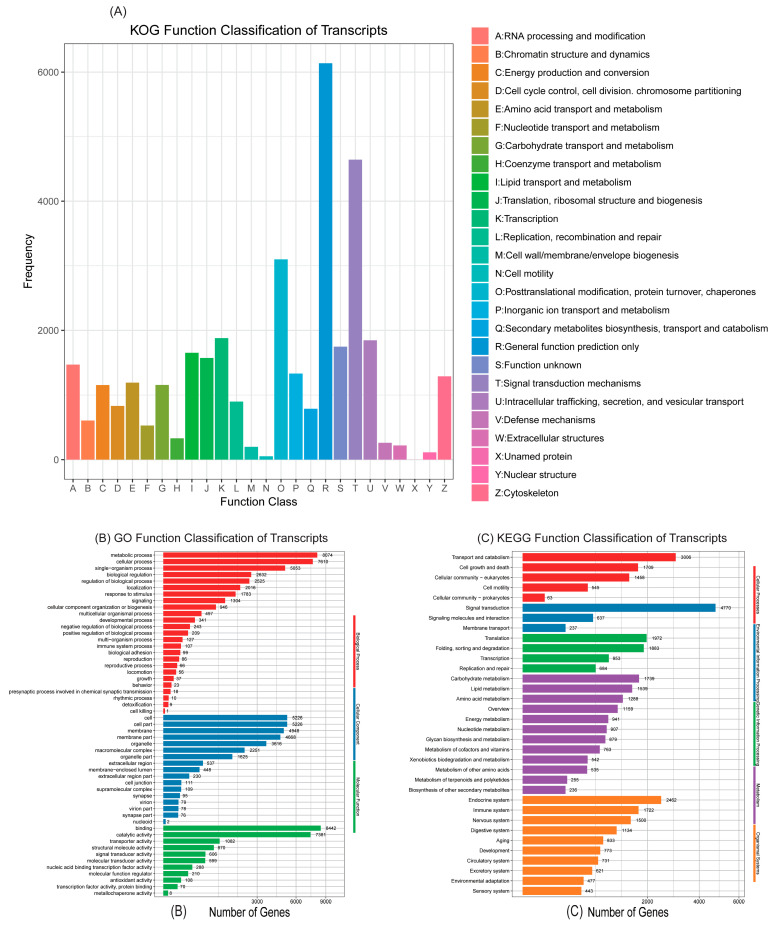
Analyses of the antennal transcriptome of *S. cerealella*. (**A**) Eukaryotic Ortholog Groups of proteins (KOG) annotation of the transcriptome. (**B**) Gene Ontology (GO) annotation of the transcriptomic sequences. (**C**) Kyoto Encyclopedia of Genes and Genomes (KEGG) pathway annotation of the transcriptome. The horizontal axis is logarithmic.

**Figure 2 insects-16-00461-f002:**
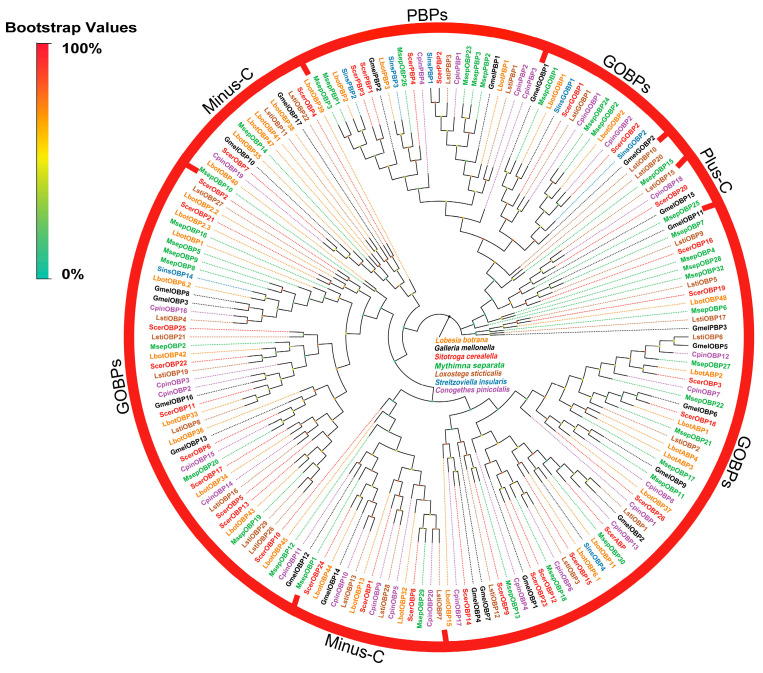
Phylogenetic tree of ScerOBPs. The phylogenetic analysis from OBPs of *S. cerealella* (ScerOBP, highlighted in red) was conducted using reference OBPs of *Streltzoviella insularis* (SinsOBP, highlighted in blue), OBPs of *Conogethes pinicolalis* (CpinOBP, highlighted in purple), OBPs of *Mythimna separata* (MsepOBP, highlighted in green), OBPs of *Galleria mellonella* (GmelOBP, highlighted in black), OBPs of *Loxostege sticticalis* (LstiOBP, highlighted in brown), and OBPs of *Lobesia botrana* (LbotOBP, highlighted in orange). The stability of the nodes was assessed by bootstrap analysis with 1000 replications. Colored dots at nodes represent bootstrap values, and branch colors indicate different species classifications.

**Figure 3 insects-16-00461-f003:**
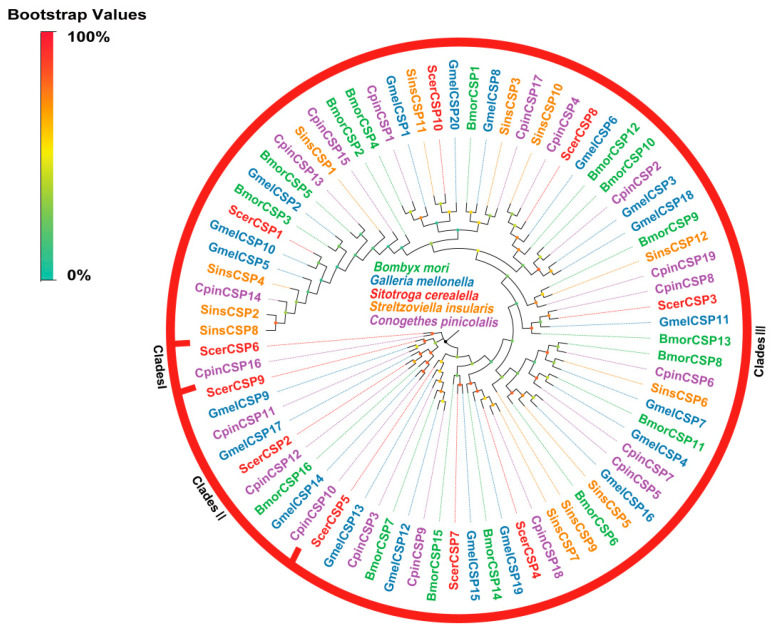
Phylogenetic tree of ScerCSPs. The phylogenetic analysis of CSPs from *S. cerealella* (ScerCSP, highlighted in red) was conducted using reference CSPs of *Galleria mellonella* (GmelCSP, highlighted in blue), CSPs of *Conogethes pinicolalis* (CpinCSP, highlighted in purple), CSPs of *Bombyx mori* (BmorCSP, highlighted in green), and CSPs of *Streltzoviella insularis* (SinsCSP, highlighted in orange). The stability of the nodes was assessed by bootstrap analysis with 1000 replications. Colored dots at nodes represent bootstrap values, and branch colors indicate different species classifications.

**Figure 4 insects-16-00461-f004:**
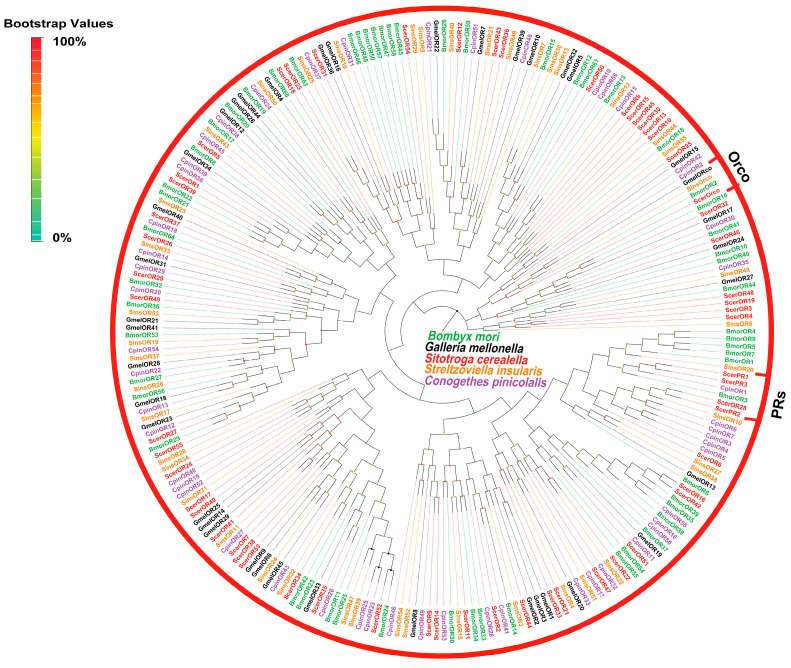
Phylogenetic tree of ScerORs. The phylogenetic analysis of ORs from *S. cerealella* (ScerOR, highlighted in red) was conducted using reference ORs of *Galleria mellonella* (GmelOR, highlighted in black), ORs of *Conogethes pinicolalis* (CpinOR, highlighted in purple), ORs of *Bombyx mori* (BmorOR, highlighted in green), and ORs of *Streltzoviella insularis* (SinsOR, highlighted in orange). The stability of the nodes was assessed by bootstrap analysis with 1000 replications. Colored dots at nodes represent bootstrap values, and branch colors indicate different species classifications.

**Figure 5 insects-16-00461-f005:**
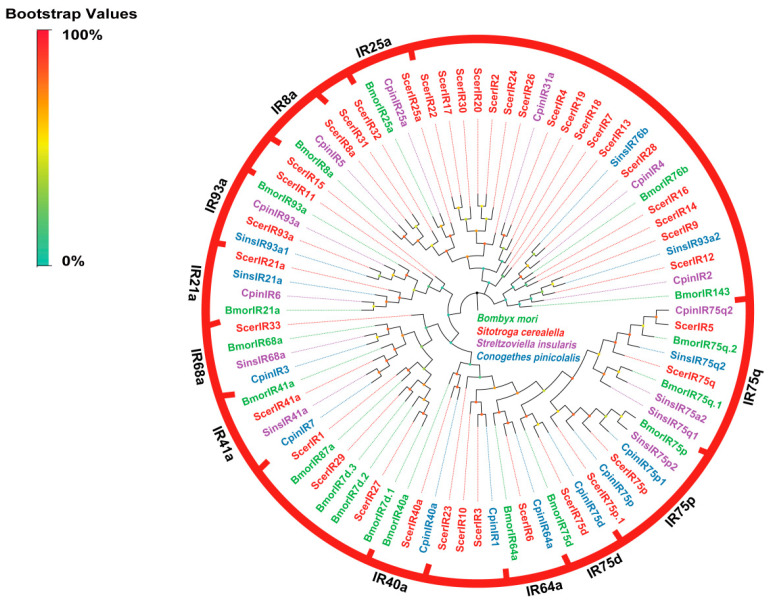
Phylogenetic tree of ScerIRs. The phylogenetic analysis of IRs from *S. cerealella* (ScerIR, highlighted in red) was conducted using reference IRs of *Conogethes pinicolalis* (CpinIR, highlighted in blue), IRs of *Bombyx mori* (BmorIR, highlighted in green), and IRs of *Streltzoviella insularis* (SinsIR, highlighted in purple). The stability of the nodes was assessed by bootstrap analysis with 1000 replications. Colored dots at nodes represent bootstrap values, and branch colors indicate different species classifications.

**Figure 6 insects-16-00461-f006:**
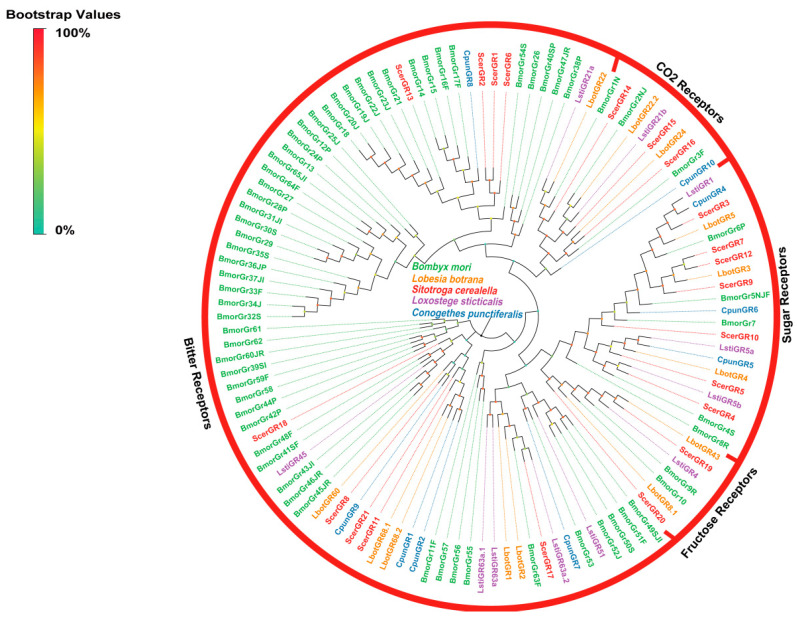
Phylogenetic tree of ScerGRs. The phylogenetic analysis of GRs from *S. cerealella* (ScerGR, highlighted in red) was conducted using reference GRs of *Conogethes punctiferalis* (CpunGR, highlighted in blue), GRs of *Bombyx mori* (BmorGR, highlighted in green), GRs of *Lobesia botrana* (LbotGR, highlighted in orange), and GRs of *Loxostege sticticalis* (LstiGR, highlighted in purple). The stability of the nodes was assessed by bootstrap analysis with 1000 replications. Colored dots at nodes represent bootstrap values, and branch colors indicate different species classifications.

**Figure 7 insects-16-00461-f007:**
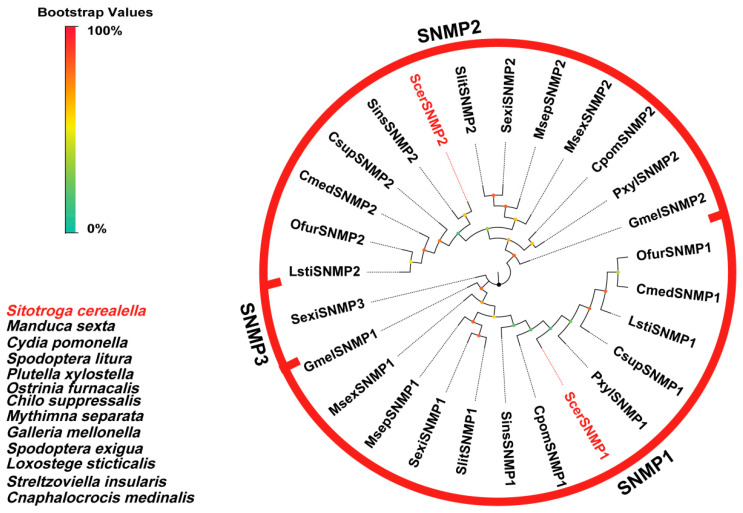
Phylogenetic tree of ScerSNMPs. The phylogenetic analysis of SNMPs from *S. cerealella* (ScerSNMP, highlighted in red) was conducted using reference SNMPs from twelve other insects. The stability of the nodes was assessed by bootstrap analysis with 1000 replications. Colored dots at nodes represent bootstrap values, and branch colors indicate different species classifications.

**Figure 8 insects-16-00461-f008:**
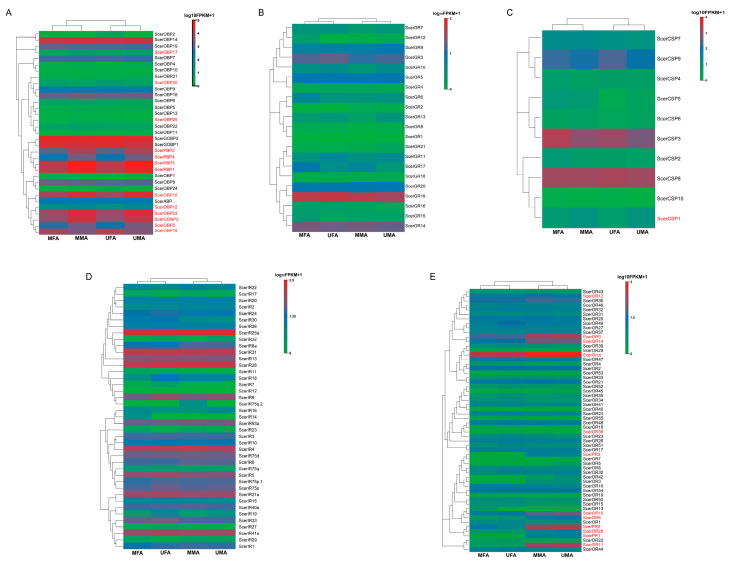
Expression levels (FPKM) of olfactory genes in mated and unmated female and male antennae are displayed behind the gene names (red indicates that there are differences in gene expression levels between male and female antennae). Expression values (FPKM) are shown on a log2 scale. (**A**): ScerOBPs, (**B**): ScerGRs, (**C**): ScerCSPs, (**D**): ScerIRs, (**E**): ScerORs; MFA: mated female antennae, UFA: unmated female antennae, MMA: mated male antennae, UMA: unmated male antennae.

**Figure 9 insects-16-00461-f009:**
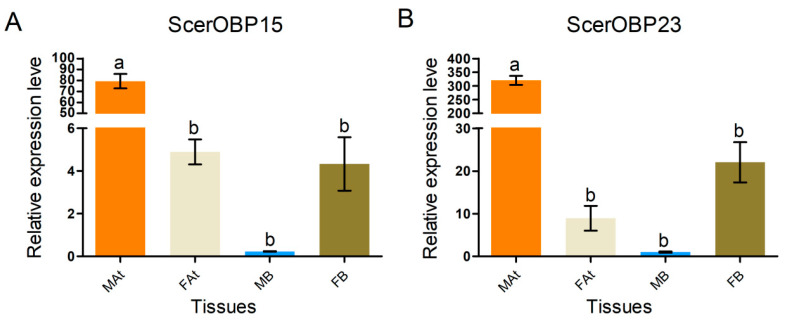
Odorant-binding proteins (OBPs) expression levels of *S. cerealella* in different chemosensory tissues. MAt: male antennae; FAt: female antennae; MB: male bodies; FB: female bodies. Standard errors are represented by the error bar. Different letters (a, b) above each bar denote significant differences (*p* < 0.05). (**A**) ScerOBP15; (**B**) ScerOBP23.

**Figure 10 insects-16-00461-f010:**
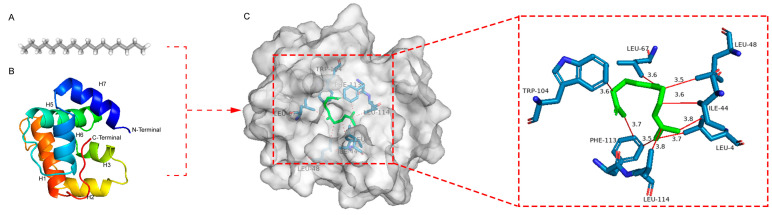
The binding mode and interaction diagram of ScerOBP15 with geranyl acetone (Hydrophobic interactions: red). (**A**) geranyl acetone; (**B**) The 3D model of the target protein ScerOBP15 based on the crystal structure of the template protein of AgamOBP1; (**C**) Three-dimensional structure of the combined model between ScerOBP15 and geranyl acetone.

**Figure 11 insects-16-00461-f011:**
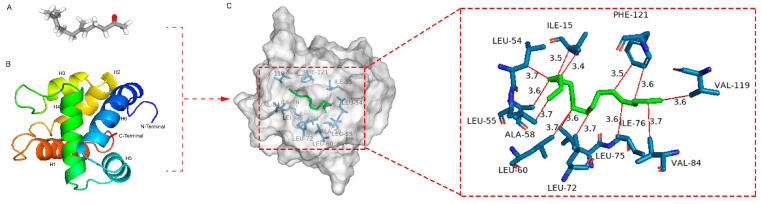
The binding mode and interaction diagram of ScerOBP23 with n-heptadecane (Hydrophobic interactions: red). (**A**) n-heptadecane; (**B**) The 3D model of the target protein ScerOBP23 based on the crystal structure of the template protein of AgamOBP1; (**C**) Three-dimensional structure of the combined model between ScerOBP23 and n-heptadecane.

**Table 1 insects-16-00461-t001:** Summary of reads and transcripts from the antennal transcriptome of *S. cerealella*.

Category	Unmated Male	Unmated Female	Mated Male	Mated Female
Clean reads (million)	25.33 ± 4.61	26.65 ± 1.95	23.75 ± 1.36	26.64 ± 1.27
Average length (bp)	150	150	150	150
Q20 (%)	98.23 ± 0.16	98.13 ± 0.16	98.23 ± 0.06	98.20 ± 0.20
Q30 (%)	92.60 ± 0.44	92.17 ± 0.23	92.53 ± 0.06	92.53 ± 0.75
GC (%)	45.07 ± 0.38	44.73 ± 1.89	44.70 ± 0.46	45.63 ± 0.70
Transcript length interval	Number of Transcripts		Percentage (%)	
<500 bp	47,663		33.48	
500–1k bp	30,599		21.49	
1k–2k bp	30,716		21.57	
>2k bp	33,404		23.46	
Total Transcripts	142,382		100.00	
Annotated Transcript	Number of Transcripts		Percentage (%)	
Annotated in NR	67,529		47.43	
Annotated in KEGG	34,591		24.29	
Annotated in SwissProt	45,867		32.21	
Annotated in GO	17,666		12.41	
Annotated in KOG	30,849		21.67	
Unknown	74,629		52.41	
Total Transcripts	142,382		100	

**Table 2 insects-16-00461-t002:** Modeling information of ScerOBP15 and ScerOBP23 in *S. cerealella*.

Name	Template	Species/Protein Name	Seq Identify	Coverage
ScerOBP15	2erb.1.A	*Anopheles gambiae*/AgamOBP1	40.37%	0.95
ScerOBP23	3ogn.1.A	*Culex quinquefasciatus*/OBP	31.93%	0.90

**Table 3 insects-16-00461-t003:** Binding energy of odor compounds and ScerOBPs.

Compounds	Molecular Formula	CAS Number	Binding Energy (kcal/mol)	
			ScerOBP15	ScerOBP23
tetradecane	C14H30	629-59-4	−5.402	−5.392
1-pentadecene	C15H30	13360-61-7	−5.308	−5.349
hexadecane	C16H34	544-76-3	−5.386	−5.293
n-heptadecane	C17H36	629-78-7	−5.685	−5.938
1-iododecane	C10H21I	2050-77-3	−4.406	−4.901
nonylaldehyde	C9H18O	124-19-6	−4.468	−4.595
capric aldehyde	C10H20O	112-31-2	−4.384	−4.608
palmitic acid	C16H32O2	57-10-3	−5.509	−5.37
oleic acid	C18H34O2	112-79-8	−6.016	−5.907
pentadecanoic acid	C15H30O2	1002-84-2	−5.721	−5.632
methylheptenone	C8H14O	110-90-3	−4.638	−4.755
geranyl acetone	C13H22O	689-67-8	−6.201	−5.811

## Data Availability

All the sequencing data generated in this study were submitted to the National Genomics Data Center under accession number: PRJNA877069.

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
