# Peer review of "Deciphering the Olfactory Mechanisms of Sitotroga cerealella Olivier (Lepidoptera: Gelechiidae): Insights from Transcriptome Analysis and Molecular Docking"

_insects, 2025, doi:10.3390/insects16050461_

Round 1

Reviewer 1 Report

Comments and Suggestions for Authors

This manuscript is fairly well written and presented; the authors explore transcriptome analysis of Sitotroga cerealella, a globally widespread pest of stored grain, with possible applications to its control. The authors should include the common name of this moth (the Angoumois grain moth) in their abstract and introduction

Also of note; sometimes chemical names in the manuscript were capitalized even when they weren’t at the beginning of a sentence (they shouldn’t be) and scientific names of insects weren’t italicized in some places. There should be consistency throughout the manuscript. This is noted in my editorial comments (see below).

Although I don’t have a background in genetics, this manuscript seems scientifically sound and well-written. I only noted a few minor editorial comments, see below:

-line 13, insert a comma after “proteins”

-line 17 and elsewhere, the H of n-Heptadecane does not need to be capitalized; the names of chemicals don’t need to be capitalized unless they are at the beginning of a sentence.

-lines 26 and 27, maybe include “BLASTx” and “qRT-PCR” in the list of abbreviations at the end. Also SOAPnuke, line 146, and FPKM line 169.

-line 20, “insect” not “insects”

-line 29, just so people without a background in genetics can understand, the molecular docking was simulated by computer software, unless I am mistaken?

-line 45, put a space after “(SNMPs),”

-line 56, omit “antenna”

-line 81, “co-localizes” not “co-localize”

-line 126, remove “from”

-line 155 “…five publicly protein databases”; do the authors mean “five publicly available protein databases”?

-line 206, remove “a”

-line 225 and elsewhere, insert a blank line after the table and before the text that comes after.

-line 228, “abundance” not “abundan”

-line 238k “antennal” not “antennal”

-line 240, maybe include KEGG in the list of abbreviations at the end of the text.

-line 286, insert “the” after “that”

-line 306 and elsewhere, shouldn’t the species names be italicized, at least to keep consistency throughout the document?

-line 313, insert a space before “Neighbor”

-line 367, remove “in” after “among”

-line 391, suggest “structures” not “structural”

-line 399, maybe “considered to be substantial at -5.00 kJ/mol…” as an alternative wording…unless I miss the meaning of the sentence?

-line 457, insert a space after “species.”

-line 463, typo after “functions”; please correct.

-line 483 maybe “not developing” instead of “did not develop” (an alternative wording that is more clear).

-line 506, implicated, not rimiplicated.

-line 517, receptor not receptors.

-line 520, do the authors mean “butter moths”? This should be made clear.

-line 524 “n-hepadecane” not “n-Heptadecan” here and elsewhere.

-linr 556, maybe insert “a” before “static” and the sentence would flow a little better.

Once these minor editorial issues are corrected (minor revision) I feel the paper is publishable.

Reviewer 2 Report

Comments and Suggestions for Authors

In this study, the authors identified 165 putative olfactory genes in Sitotroga cerealella based on the antennal transcriptome. These genes include odorant binding proteins, chemosensory proteins, odorant receptors, ionotropic receptors, gustatory receptors, and sensory neuron membrane proteins. Subsequently, through qRT - PCR, the authors demonstrated that ScerOBP15 and ScerOBP23 are highly expressed in male antennae. Molecular docking assays further indicated that these two proteins may be involved in host localization. This research is both fascinating and valuable for elucidating the olfactory mechanism of Sitotroga cerealella.

However, the study has certain limitations. It remains unclear why the authors only conducted docking analyses on two OBPs, with no explanation or discussion provided for this choice. Additionally, there is a lack of exploration regarding why the male - biased genes are solely associated with binding host odorants. To enhance the comprehensiveness of this study, it would be beneficial if the authors could perform molecular docking for all identified olfactory proteins. Moreover, validating their functions through electroantennography assays would further strengthen the research findings.

Reviewer 3 Report

Comments and Suggestions for Authors

In this work, the authors explore antennal transcriptomes of an invasive pest Sitotroga cerealella.

in order to uncover the diversity of olfaction-related genes and dive quite deep into the matter by analyzing their phylogenies and performing molecular docking. I’d like to commend the authors for making their analysis quite reproducible by making their data available and providing accession numbers for genes used in blast analysis. The study is sound in general, but in my opinion it has a room for improvement to make a larger impact.

Major questions / suggestions:

1. (to Introduction): the description of different odorant-related groups is very confusing.

For example, L62-63 “Similar to OBPs, CSPs are small soluble proteins”: and what is the difference?

Moreover, L74-45 “molecular structure similar to that of ionotropic glutamate receptors”: which is?

As this is important in the context of results, it would be great to make this description clearer and easy to follow. I would suggest making a table of all the groups with their properties important for the interpretation of results, but this is of course up to the authors.

2. Reproducibility.

The data published alongside the article, which is great! However, I did not fully understand the structure of the data.

https://www.ncbi.nlm.nih.gov/biosample?Db=biosample&DbFrom=bioproject&Cmd=Link&LinkName=bioproject_biosample&LinkReadableName=BioSample&ordinalpos=1&IdsFromResult=877069

First, I see two BioSamples, while there should be at least four. What happened? Do these runs contain mixed samples? How could those be separated?

Second, it is planned to make the assembly available as well?

3. How were the phylogenetic trees rooted? couldn’t find such information in the methods. In theory, midpoint rooting (if this was used) with the NJ method could lead to artifacts. In general, outgroup-based rooting is more reliable—of course, if suitable outgroups exist for a particular analysis.

4. Fig. 2

- Why is the “Minus-C” group mentioned twice in unrelated tree branches? Are “Minus-C” proteins two unrelated groups? This question is perhaps related to question #1 to the Introduction, which also highlights the importance of providing a classification of olfactory-related proteins.

In addition, what are the groups without labels?

Minor suggestions:

5. L40 “biological behaviors” sounds excessive (can a behaviour be non-biological?)

6. L126 “body devoid of”: of antennae? thereof?

7. Fig. 1: “antannal” should be “antennal”

L238 “antannal” should be “antennal”

8. Fig. 2 could be enlarged without trouble (too much white space, too little text). The same goes for Fig. 4.

9. Please consider adding some information to Figure 2-4 legends that bootstrap values are color-coded as dots at the nodes, while the color of the branches codes for species. It took me a long time to realize.

10. Please consider adding to Figure 1 legend some information that the horizontal axis is logarithmic. It would also be helpful to enlarge the fonts wherever possible.

Round 2

Reviewer 3 Report

Comments and Suggestions for Authors

The authors did a great job in revising the manuscript, and I’d like to thank them for honestly taking the critiques raised and questions asked. I have some minor comments in response to the authors’ comments. Please find them below.

Re “About the suggestion to create a classification table: We truly appreciate this constructive suggestion. However, after careful consideration, we decided to maintain the current organization because: (1) The introduction aims to briefly introduce all olfactory-related proteins (OBPs, CSPs, ORs, IRs, GRs, and SNMPs) as background; (2) The results section provides detailed descriptions and discussions of each protein family with comprehensive analysis. We believe this flow better serves readers by first introducing concepts then providing in-depth analysis later.”: I totally do not insist on including the table into the manuscript, it was just a suggestion and totally up to the authors. I agree with the authors it is not necessary for the Introduction section in an experimental manuscript.

Re “We acknowledge that the current NCBI display presents the data grouped by sex only, which may have caused confusion. The complete raw data with detailed grouping information is

thoroughly documented in: Section 3.1 "Transcriptome sequencing, Assembly, and Annotation"

of our manuscript and Table 1 ("Summary of reads and transcripts from the antennal transcriptome of S. cerealella") For details, see page 5, paragraph 4, line 222 and 233 of the

revised manuscript.”

Thanks for the explanation! Now I definitely understand more but I’m still lost about how I could reproduce the analysis if needed. I guess that some barcode adapters were ligated before pooling the sample for sequencing and then used to split the raw reads into groups of reads corresponding to individual samples. If my hypothesis is correct, please consider including the sequences of these barcode adapters into supplementary materials.

Re “The remaining unmarked clusters represent conventional OBPs”: please consider adding this information into the figure caption as well in case some of the readers have the same questions as I did.
